# The Solubility and Structures of Porcine Myofibrillar Proteins under Low-Salt Processing Conditions as Affected by the Presence of L-Lysine

**DOI:** 10.3390/foods11060855

**Published:** 2022-03-17

**Authors:** Xiuping Li, Wenhui Wang, Shouyin Wang, Yuqing Shen, Jinfeng Pan, Xiuping Dong, Shengjie Li

**Affiliations:** 1School of Food Science and Technology, Dalian Polytechnic University, Ganjingzi #1, Dalian 116304, China; lixiuping1176@163.com (X.L.); fumei1998@126.com (W.W.); shouyin_wang@126.com (S.W.); shenyuqing0715@163.com (Y.S.); panjf@dlpu.edu.cn (J.P.); dongxp@dlpu.edu.cn (X.D.); 2National Engineering Research Center of Seafood, Dalian Polytechnic University, Ganjingzi #1, Dalian 116304, China

**Keywords:** myofibrillar proteins, L-lysine, solubility, conformation, low salt

## Abstract

This study aimed to investigate the presence of L-lysine (Lys) on the solubility and structures of myofibrillar proteins (MFPs) at different ionic strengths. Porcine MFPs were incubated at 4 °C with various levels of ionic strengths (0.15, 0.3, or 0.6 M NaCl) with or without the presence of 20 or 40 mM Lys. After 24 h of incubation, MFP solubility and turbidity were determined, and the particle size distribution, circular dichroism spectra, and intrinsic tryptophan fluorescence of MFP were analyzed to obtain their secondary and tertiary structure. Results showed that the solubilization effects of Lys on MFPs are dependent on the ionic strength. Particularly, the presence of Lys could improve MFP solubility at 0.3 M, which resembles salt-reducing processing conditions. Concomitantly, the secondary and tertiary structures were observed to change as a result of the varying ionic strengths and the addition of Lys, including myofibril swelling, dissociation of myosin filaments, uncoiling of α-helix, and unfolding of the tertiary structure. The possible mechanisms underlying the solubilization effects of Lys on MFPs at low ionic strengths are discussed from the perspective of protein structural changes.

## 1. Introduction

Muscle proteins can be categorized into three groups on the basis of their solubility characteristics: sarcoplasmic proteins, myofibrillar proteins, and stromal proteins [1]. Myofibrillar proteins (MFPs) refer to proteins that are soluble in concentrated salt solutions, which accounts for 50–55% of the total muscle proteins. It has been generally accepted that MFPs are largely responsible for the functional properties of muscle proteins, including water binding, gelation, and emulsification, and these functionalities are the crucial factors contributing to the palatability or sensory perceptions (tenderness, mouthfeel, juiciness, etc.) of processed meat products [2]. In particular, MFPs’ functional properties are critically dependent on their solubilization or extraction during meat processing. This is because MFPs extracted from myofibrils become gelatinized upon heating, which is responsible for the formation of a three-dimensional network structure [3]. During meat processing, the solubilization of MFPs is normally dictated by the amount of salt added. In general, the addition of 2.0–3.0% NaCl is typically required for the adequate solubilization of MFPs, and these percentages of NaCl are equivalent to ionic strengths of 0.47–0.68 M [4]. Considering that increased sodium consumption is significantly associated with an increased risk of cardiovascular disease, great efforts have been made to reduce the salt contents in different meat products; however, a number of challenges still remain [5]. Particularly, because the extraction of MFPs in the processing of salt-reduced meat products is negatively influenced when less salt is incorporated in the formulation, the textural properties of meat products could be impaired [6]. In this context, to develop low-salt meat products, it is of critical importance to increase the solubility of MFPs under low-salt processing conditions.

In recent years, the effects of different basic amino acids on the solubilization of myosin (one of the most important MFPs) have been extensively studied, such as L-lysine (Lys) [7], L-arginine (Arg) [8], and L-histidine (His) [9,10,11]. At the very low ionic strength (1 mM KCl, pH 7.5), more than 80% of the myosin was observed to be solubilized in the presence of 5 mM His [10]. However, the solubilization effects of His were found to be marginal at the ionic strength of 0.15 M KCl, with negligible effects at 0.6 M KCl [9]. At the ionic strength of 0.15 or 0.2 M NaCl, salient effects of 50 mM Arg or Lys on the solubilization of myosin were observed, whereas their solubilization effects were minimal at relatively low ionic strengths (0.05–0.1 M) and negligible at high ionic strengths (0.25–0.3 M) [8]. On the other hand, 5 mM of Lys was able to increase the solubility of myosin at the different ionic strengths (1 mM, 0.3 M, and 0.6 M) [7]. Based on this evidence, the solubilization effects of these basic amino acids might be dependent on the ionic strengths, but this hypothesis has yet to be tested. Moreover, although the solubilization effects of these basic amino acids on myosin have been extensively examined, little is known about their effects on MFPs, which is a more complex subject but better resembles the in situ situation in meat processing. It is noteworthy that the addition of 0.8% Lys could improve the water-holding capacity and textural properties of salt-reduced ham [6]. Given that the ionic strength of 0.3 M is more relevant to the processing condition of reduced-salt meat products, whether the presence of Lys could improve the solubilization of MFPs at such an ionic strength remains unclear.

It has long been recognized that the functional properties of muscle proteins are highly associated with their structures [2]. The low solubility of MFP in solutions of physiological and low ionic strength has been ascribed to the polymerization of myosin (one of the major MFP) to form a filamentous structure [12]. Moreover, the increased solubility of myosin in low salt solutions in the presence of 5 mM His has been shown to be accompanied by structural changes of myosin [9]. Circular dichroism (CD) analysis revealed that the presence of His or Lys exhibited significant influences on the secondary structures of myosin regardless of the ionic strength, which was manifested as a decreased content of α-helix [7]. Therefore, to elucidate the solubilization mechanism of Lys, the structural changes in MFP in relation to the presence of Lys need to be clarified.

The present study was designed to investigate the effects of Lys on the solubility and conformations of MFPs at different ionic strengths (0.15, 0.3, and 0.6 M).

## 2. Materials and Methods

### 2.1. Materials

L-Lysine were purchased from Beijing Solarbio Science & Technology Co., Ltd. (Beijing, China), with other common chemicals phased from Shanghai Aladdin Biochemical Technology Co., Ltd. (Shanghai, China). All chemicals used were of reagent grade and were used as received, including NaCl, NaH_2_PO_4_, Na_2_HPO_4_, MgCl_2_, EGTA, etc.

Porcine longissimus dorsi muscles were obtained from a local slaughterhouse within 48 h postmortem and transferred to the lab on ice. After the removal of visible fat and connective tissues, porcine meat was cut into small cubes (1 cm × 1 cm × 1 cm), which were then portioned and stored at −40 °C until use.

### 2.2. MFP Extraction

MFPs were extracted from porcine longissimus dorsi muscle according to Huff-Lonergan et al. [13], with minor modifications. Briefly, following thawing overnight at 4 °C, 50 g of meat sample were minced and then homogenized in 200 mL of extraction buffer (0.1 M NaCl, 10 mM sodium phosphate, 2 mM MgCl_2,_ 1 mM EGTA, pH 7.0) at 10,000 rpm for 30 s using an Ultra-Turrax T25 homogenizer (IKA Labortechnik, Staufen, Germany), and the homogenate was centrifuged at 2000× *g* for 15 min (4 °C). The supernatant was discarded, and the obtained pellet was washed twice with the extraction buffer. Afterwards, the protein pellets were then washed 3 times with 0.1 M NaCl. In the third circle of the washing step, protein suspensions were filtered by gauze before centrifugation to remove connective tissues. Finally, the suspensions were centrifuged at 2000× *g* for 15 min (4 °C), and the obtained pellets were used immediately for the preparation of MFP suspensions with assorted levels of ionic strength and Lys.

### 2.3. Preparation of MFP Suspensions with Assorted Levels of Ionic Strength and Lys

The final pellets obtained in Section 2.2 were portioned and mixed with one of the following incubation buffers: (1) 0.15 M NaCl, 10 mM sodium phosphate, pH 6.0; (2) 0.15 M NaCl, 20 mM Lys, 10 mM sodium phosphate, pH 6.0; (3) 0.15 M NaCl, 40 mM Lys, 10 mM sodium phosphate, pH 6.0; (4) 0.3 M NaCl, 10 mM sodium phosphate, pH 6.0; (5) 0.3 M NaCl, 20 mM Lys, 10 mM sodium phosphate, pH 6.0; (6) 0.3 M NaCl, 40 mM Lys, 10 mM sodium phosphate, pH 6.0; (7) 0.6 M NaCl, 10 mM sodium phosphate, pH 6.0; (8) 0.6 M NaCl, 20 mM Lys, 10 mM sodium phosphate, pH 6.0; (9) 0.6 M NaCl, 40 mM Lys, 10 mM sodium phosphate, pH 6.0. After mixing thoroughly, the protein concentrations of the assorted MFP suspensions were measured and then adjusted to the same level of 10 mg/mL. Afterwards, the pH of these different MFP suspensions was measured and adjusted to the same level to exclude the pH effect on MFP solubilization. MFP suspensions were incubated at 4 °C for 24 h, followed by various analyses as described below.

### 2.4. MFP Solubility

The solubility of MFP was determined as reported by Li et al. [14] with slight modifications. Briefly, MFP solutions was centrifuged at 10,000× *g* for 15 min at 4 °C, and the protein concentration of the supernatant was then determined by a BCA assay kit (Beijing Biolab Co. Ltd., Beijing, China). MFP solubility was expressed as the ratio of protein concentration of the supernatant to that of the MFP suspension before centrifugation.

### 2.5. MFP Turbidity

The turbidity was measured as reported previously [15]. First, 1 mg/mL of MFP solutions were obtained by diluting the original MFP suspension with the corresponding incubation buffers, which were transferred to a quartz cuvette. The absorbance at 660 nm was obtained using a spectrophotometer (Lambda 35, Perkin Elmer, Waltham, MA, USA), and the arbitrary absorbance was regarded as the protein turbidity.

### 2.6. Measurement of the Particle Size

The particle size distribution was measured using a Zetasizer 3000HSA (Malvern Panalytical Ltd., Malvern, UK) as reported previously [15]. In brief, diluted MFP solutions (1 mg/mL) were obtained with respective incubation buffers, which were then scanned at an excitation wavelength of 633 nm with a detector angle of 173°. 

### 2.7. Measurement of Circular Dichroism (CD) Spectra

The CD spectrum was measured following a previously reported method with modifications [11]. The diluted MFP solutions (0.3 mg/mL) with the respective incubation buffer were transferred to a quartz cell with a 0.1-cm path length. CD signals were recorded in the wavelength range between 260 and 200 nm, and the scanning rate was set at 100 nm/min using a Jasco J-1500 CD spectrometer (Jasco Co. Ltd., Tokyo, Japan). Before the measurements of the samples from the same treatment, a baseline was scanned using corresponding incubation buffers, and MFP samples of the same treatment were then scanned with the identical settings. After measurement, the mean residue molar ellipticity was calculated using the average amino acid residue weight of 110 Daltons. Afterwards, the percentages of the α-helix secondary structure were then calculated using the protein secondary structure estimation program (Yang’s method) provided with the Jasco J-1500 CD spectrometer. 

### 2.8. Measurement of Intrinsic Tryptophan Fluorescence

The intrinsic tryptophan fluorescence of the MFP solution was measured according to the method of Chen et al. [11] with minor modifications. Briefly, the protein solution was diluted to 1 mg/mL with corresponding incubation buffer. Afterwards, the fluorescence spectra were recorded in the emission wavelength range from 300 to 400 nm with an excitation wavelength of 280 nm using a fluorescence spectrophotometer (F-2700, Hitachi High-Tech Corporation, Tokyo, Japan).

### 2.9. Statistical Analysis

All data are presented as the mean ± standard deviation (SD) of three independent experiments. Two-way ANOVA (analysis of variance) were performed using R (version 4.1.2, R Foundation for Statistical Computing, Vienna, Austria) to analyze the effects of ionic strength, Lys addition, and their interaction. The least square means of different treatments were compared using the turkey test at the level of *p* < 0.05. Log-transformation of the solubility data was conducted since the original data violated the assumption of equal variance.

## 3. Results

### 3.1. Solubility

As illustrated in Figure 1, when Lys was not present, treatment with an ionic strength of 0.15 M had the lowest MFP solubility (8.1%), whereas treatment with 0.6 M had the highest solubility (95.9%). Meanwhile, there was slight increase in the treatment with an ionic strength of 0.3 M (9.7%) in comparison with the treatment of 0.15 M. At the ionic strength of 0.15 M, the presence of 20 mM Lys significantly increased the solubility of MFPs compared to the control group (17.1% vs. 8.1%), and a further increase was observed in the group with 40 mM Lys (30.8%). Similar solubilization effects of the Lys concentration were observed at the ionic strength of 0.3 M, with the highest value in the treatment with 40 mM Lys (39.3%). However, at the ionic strength of 0.6 M, neither 20 mM Lys nor 40 mM Lys showed any solubilization effects on MFPs compared to the control. In the control group, MFP was almost totally solubilized at the ionic strength of 0.6 without Lys addition (95.9%).

### 3.2. Turbidity

The solubility of MFP is closely related to the protein aggregation, and the solubility decreases as protein forms aggregates. In this regard, solubilized proteins can exist in a colloidal form and dispersed in solution rather than as separated molecules in true solution that does not scatter light. To assess the aggregation of MFP in relation to the ionic strength and the presence of Lys, the turbidities of MFP solutions were measured (Figure 2). At the ionic strength of 0.15 M, the turbidity of MFP solutions decreased significantly in the treatment with 20 mM Lys compared to the control, whereas no further increase was observed in the treatment with 40 mM Lys. Similar effects of Lys addition were observed at the ionic strength of 0.3 M, where the turbidity for groups with 20 or 40 mM Lys was much smaller than the control. When the ionic strength was 0.6 M, however, the groups with various Lys had similar levels of turbidity to that of the control. In addition, the turbidity of the treatments with 20 or 40 mM Lys in the presence of 0.3 M NaCl was only slightly higher than that in the presence of 0.6 M NaCl without Lys.

### 3.3. Particle Size Distribution

The aggregation of MFP is also manifested by the particle size of the protein solutions (Figure 3). At the ionic strength of 0.15 M, a single peak at around 600–1000 nm was observed for all the treatments regardless of Lys addition. Nevertheless, compared to the control, the peak position for the treatment with 40 mM Lys shifted towards a longer wavelength to 1000 nm. At the ionic strength of 0.3 M, the control group exhibited 1 single peak at 1000 nm, whereas the group with 20 mM Lys presented 2 peaks at 700 and 3000 nm, respectively, with the latter accounting for a larger proportion. Similarly, the group with 40 mM Lys also had 2 peaks, located at 400 and 3000 nm, respectively, and the latter peak was the main component. At the ionic strength of 0.6 M, there were 3 peaks present in the control group, which were located at around 60, 400, and 1000 nm, where the latter 2 peaks were not seperate from each other. Peaks at 60 and 400 nm were also present in the treatments with 20 or 40 mM Lys, whereas the third peak for the treatments with Lys shifted towards a longer wavelength, where the more Lys was added, the longer the peak wavelength was. 

### 3.4. Protein Secondary Structure

CD is an excellent tool to analyze the secondary structures of proteins, consisting of α-helix, β-sheet, β-turn, and random coil [16]. As shown in Figure 4, at the ionic strength of 0.15 M, the CD spectra of all 3 treatments (control, 20 mM Lys, and 40 mM Lys) appeared to be symmetric and had a single negative band at 227 nm. Moreover, the presence of Lys increased the peak value of the mean residue molar ellipticity in comparison with the control, indicating an increase in the relative contents of α-helix in the treatments with Lys. At the ionic strength of 0.3 M, the spectrum of the control was asymmetric, and had a negative band round the wavelength of 227 nm. However, 2 negative bands at 212 and 222 nm were observed in both treatments with Lys. Furthermore, compared to the control at 0.15 M, the control treatment at the ionic strength of 0.3 M increased the peak value of the mean residue molar ellipticity, and the presence of Lys led to further increases. When the ionic strength increased to 0.6 M, 2 negative bands at 212 and 222 nm were presented in the control and 20 mM Lys groups, whereas a positive band at 206 nm besides these 2 negative bands was also found in the 40 mM Lys group, which probably originated from the absorbance of Lys molecules (data not shown). More importantly, a further increase in the peak value was observed in the control compared to the same treatment at 0.3 M. However, at the ionic strength of 0.6 M, the presence of Lys decreased the peak values of the mean residue molar ellipticity in comparison with the control, indicating the Lys addition decreased the contents of α-helix in MFP at the ionic strength of 0.6 M. 

Based on the CD spectra, the relative contents of the α-helix were calculated in all the treatments (Figure 5). At the ionic strength of 0.15, the treatment with 40 mM Lys was observed to significantly increase the α-helix contents compared to the control, whereas the treatment with 20 mM Lys showed no significant increase. In contrast, treatments with 20 or 40 mM Lys resulted in higher levels of α-helix than the control, but a lower α-helix content was observed for the treatment with 40 mM Lys than the 20 mM treatment. When the ionic strength was 0.6 M, compared to the control, a similar level of α-helix was found in the treatment with 20 mM Lys, but a significant lower α-helix content was shown for the 40 mM Lys treatment. In addition, when the different control groups were compared, it was found that the α-helix contents increased with the ionic strength with the highest value in the group of 0.6 M. To further study the effects of Lys on the unfolding of the α-helical coiled-coil myosin tail, MFP solutions were clarified by centrifugation at 10,000× *g* for 15 min and the supernatants were then analyzed using a CD spectrophotometer. It was found that the presence of Lys resulted in the loss of α-helix regardless of the ionic strengths (data not shown), indicating Lys could lead to the unfolding of α-helix.

### 3.5. Intrinsic Tryptophan of Fluorescence 

The intrinsic fluorescence of tryptophan has been commonly employed for the detection of tertiary structural changes of proteins because the chemical structure of tryptophan has an aromatic ring, which is responsible for the natural fluorescence emitted at 350 nm when it is excited at 280 nm [17]. As illustrated in Figure 6, the presence of Lys showed notable quenching effects on the tryptophan fluorescence regardless of the ionic strength, and the fluorescence intensity decreased with increasing Lys concentrations at the same level of ionic strength, especially at 0.15 and 0.3 M. Meanwhile, a slight red shift of the peak wavelength was observed as a result of the Lys addition. In addition, when no Lys was present, the group with 0.3 or 0.6 M ionic strength had higher fluorescence intensity than the group with 0.15 M.

## 4. Discussion

The solubility of MFP is of great importance for the manufacture of various comminuted, reconstructed, and formed meat products as MFP’s functionalities, including water binding, gelation, and emulsification, are achieved only when the proteins are in a highly soluble state. However, solubilization of MFP generally requires a relatively high ionic strength, which occurs as a result of comminution and blending of meat in the presence of adequate amounts of salt [2]. In this regard, the textural properties of low-salt meat products can be impaired when less salt is added as less MFPs are extracted. Therefore, to develop low-salt meat products, it is of vital importance to increase MFPs’ solubility under processing conditions with such low ionic strengths. For this purpose, the application of basic amino acids, including Lys, Arg, and His, has attracted considerable attention from meat scientists [6,7,8,9,10,11]. However, most of the studies conducted so far have focused on myosin at very low ionic strengths that do not resemble those used in meat processing. MFPs are the structural proteins that make up myofibrils, which is a more complex subject than myosin. On the one hand, MFPs comprise not only myosin but also many other proteins, such as actin, tropomyosin, etc., which may behave differently to myosin in response to the ionic strength. On the other hand, even more importantly, MFP contains some structural proteins that restrict protein extraction, such as actomyosin cross-bridges and M- and Z-line components in myofibrils [18]. Therefore, it is of interest to understand whether these basic amino acids possess the ability to solubilize MFP under conditions similar to the processing of low-salt meat products. Considering an ionic strength of 0.6 M NaCl has been generally accepted as an in situ condition resembling normal meat processing (Liu et al., 2000), an ionic strength of 0.3 M NaCl was considered as a salt-reducing condition in this study. Meanwhile, knowledge of the MFP structure and conformation will provide insight into the underlying mechanism of the solubilization ability of these amino acids.

### 4.1. Solubilization of MFP under Different Ionic Strengths in the Presence of Lys

The solubility experiments of this work demonstrated that the solubilization effects of Lys on MFP are dependent on the ionic strength, that is, at low ionic strengths (0.15 or 0.3 M), the addition of Lys could significantly improve MFP’s solubility, whereas Lys’ solubilization effects were diminished at a high ionic strength of 0.6 M (Figure 1). A imilar ionic strength-dependent mode of Lys on myosin solubility has been reported previously, where salient solubilization effects were observed at low ionic strengths of 0.15 or 0.2 M with negligible effects at high ionic strengths between 0.25 and 0.3 M [8]. In contrast, 5 mM of Lys was able to increase the solubility of myosin at the different ionic strengths (1 mM, 0.3 M, and 0.6 M) [7]. The differences might be attributed to the different protein subjects applied in this study. In parallel with the solubility results, the turbidity results in the present study further strengthen the conclusion that the effects of Lys on MFPs’ solubility are dependent on the ionic strength. Specifically, the presence of Lys could inhibit the aggregation of MFPs at the low ionic strength, but this effect of Lys became negligible at the high ionic strength. The low solubility of MFP at the low ionic strength is largely due to the polymerization of myosin into filaments under such conditions, whereas elevated salt concentrations can facilitate the dissociation of myosin filaments into monomer myosin, thus leading to an increased solubility of MFP [19]. Furthermore, the actomyosin cross-bridges, the M- and Z-lines of the myofibrils, confer major constraints on myosin extraction [18]. In meat processing, when meat is minced and blended with 2–3% salt, MFP extractions occur as muscle fibers and proteins undergo considerable structural changes due to electrostatic interactions between proteins and both sodium and chloride ions, for instance, (a) swelling of myofibrils (b) depolymerization of myofilaments, and (c) dissociation of actin from myosin or actomyosin from myofibrils leading to the extraction of myosin, actin, the actomyosin complex, and a number of other myofibrillar components [2]. As a result, a possible explanation for the increased solubility of MFP in salt solutions would be that in addition to those individual proteins that were truly soluble in the extract, some myofibril fragments, which may be somewhat hydrated, become less able to form sediment (that is, more “soluble”) under the mild centrifugation conditions employed in the solubility test [20]. In these regards, the disassembly of myosin filaments and the later dissociation of actomyosin determine the extraction extent of myosin from myofibrils, namely MFP solubility. When it comes to the solubilization effect of Lys, it has been reported that the presence of Lys might inhibit myosin aggregation via its interaction with acidic amino acid residues of myosin molecules, thus resulting in the increase in myosin solubility [21]. The electrostatic interactions between the positive and negative charged clusters on the surface of the myosin rods are essential for the filament formation [22]. Considering that Lys molecules are positively charged while myosin is negatively charged at pH 6.0, it could be deduced that the possible occurrence of electrostatic interactions between charged Lys and myosin may disrupt the forces required for the assembly of myosin filaments. On the other hand, as Lys belongs to the group of chaotropic ions, the accumulation of Lys in the protein–water interphase can make proteins more hydrophilic, thus contributing to the increased dispersion ability (namely, solubility) of MFP proteins in salt solutions [23]. Moreover, the present study further suggested that the positive solubilization effects of Lys also apply for MFP at low ionic strengths, in particular at an ionic strength of 0.3 M, which imitates the salt-reducing conditions in meat processing. In addition, it should be pointed out that although the presence of 40 mM Lys significantly improved MFPs’ solubility at 0.3 M NaCl, its solubility was less than half of the control with 0.6 M NaCl (39.3% vs. 95.9%, Figure 1), which suggested that the extent of MFP extraction in the salt-reducing processing conditions might still be impaired despite the presence of Lys. Therefore, our results also imply that the incorporation of other materials or additives or techniques that physically disrupt myofibrils and facilitate MFP extraction (for example, ultrasonic treatment) might be required to obtain adequate amounts of extracted MFPs in the production of salt-reduced products.

### 4.2. Structural Changes of MFPs under Different Ionic Strengths in the Presence of Lys

As the solubility of MFPs is critically dependent on their structure and conformation in salt solutions, our intent herein was to elucidate the possible mechanism of the solubilizing effects of Lys. For this purpose, assorted methods were employed to detect the tertiary and secondary changes of MFP proteins, for instance, particle size distribution, CD spectra, and intrinsic tryptophan fluorescence, which have been widely used as indicators for protein structures. As presented, changes in the MFP solubility were accompanied by marked structural changes in MFP as affected by both the ionic strength and Lys addition. Possible mechanisms in relation to the Lys solubilization effects involve myofibril swelling, dissociation of myosin filaments, uncoiling of α-helix, and unfolding of the tertiary structure.

#### 4.2.1. Swelling of Myofibrils

In meat processing, the extracted MFP can be present in different forms depending on the ionic strength. At salt concentrations below 0.3 M NaCl, MFP exists in the form of intact myofibrils, due to the polymerization of myosin, while at salt concentrations higher than 0.5 M, extracted proteins (such as actomyosin, myosin, and actin) and swollen or partially unraveled myofibril remnants will coexist [14,20]. This statement agrees with the observations from the particle size experiments in the current study (Figure 3). As illustrated, one single peak between 600 and 1000 nm appeared in MFP at the 0.15 M ionic strength, and several peaks with assorted particle sizes were present in MFP at 0.3 or 0.6 M. In particular, at these ionic strengths, peaks with a larger particle size in comparison to the control were observed in the treatments with Lys, indicating the presence of swollen myofibrils, whereas peaks with a smaller particle size from these treatments indicated the presence of extracted protein or partially unraveled myofibril remnants. Furthermore, the particle size experiments revealed that MFPs tended to swell with increased ionic strengths, and the addition of Lys enabled further swelling. It has long been recognized that myofibrils can swell to about twice their original volume in 0.6 M salt solutions, and this is because Cl^−^ anions bind selectively to the myofilaments and increase the electrostatic repulsive force between them [18]. However, the underlying mechanism regarding how Lys makes myofibrils swell remains unclear, since Lys is positively charged under the pH employed here, which would counteract the repulsive force from the Cl^−^ anions. Although the mechanism cannot be explained so far, to some extent the positive effects of Lys on myofibril swelling might contribute to the increased solubility of MFP. The possible explanation is that as myofibrils swell, the overall density of the myofibrils decreases, which would possibly make MFP become less able to form sediment during mild centrifugation.

#### 4.2.2. Dissociation of Myosin Filaments and Uncoiling of α-Helix

The α-helix conformation of MFP is predominantly determined by the α-helix structure of myosin since myosin is the single most abundant protein in MFP [1]. Structurally, a myosin molecule consists of 2 globular heads and a rod-like tail, and the tail portion of myosin contains almost 100% α-helix, whereas the head has less than 50% α-helix [24]. The coiled coil of the myosin rod has a typical seven-residue heptad repeat, making it highly charged on the myosin coil surface, which is believed to dictate its arrangement into thick filaments below physiological ionic strengths and the dissociation of thick filaments into myosin monomers at high ionic strengths [17]. In the current study, CD analysis revealed that the addition of Lys increased the relative contents of α-helix at the 0.15 and 0.3 M ionic strengths compared to the control (Figure 4 and Figure 5), and the most likely explanation, also supported by the turbidity data, is that the presence of Lys could lead to the dissociation of myosin filaments and thus more myosin molecules are dissolved as monomers such that they become optically active in CD analysis. The role of Lys played in the dissociation of myosin filaments might originate from the possible electrostatic interactions between oppositely charged myosin and Lys and the chaotropic effects of Lys as mentioned above. More interestingly, at the 0.3 M ionic strength, reduced contents of α-helix were observed in the group with 40 mM Lys in comparison with the 20 mM Lys group, which suggested the uncoiling of the helical structure of myosin, and a similar phenomenon was also observed at the 0.6 M ionic strength. In agreement with this, Guo et al. [7] reported that the presence of 5 mM Lys decreased the α-helix contents at the ionic strength of 0.6 M NaCl, whereas similar effects were also observed at 1 mM and 0.15 M, which was different from our study. In contrast, no distinguishable changes in the CD spectra of myosin were observed in the presence of 50 mM Lys with 50–300 mM NaCl [8]. Lys was found to be optically active below 230 nm (data not shown), which has also been reported previously [25]. The lack of an consistency regarding the Lys effects on the CD spectra at low ionic strengths between the current study and the previous studies might originate from the optical activity of Lys, and baseline corrections were conducted based on the respective incubation buffers in the present study. The α-helix structure is generally stabilized by the formation of the intramolecular hydrogen bonds between the NH and CO groups of the polypeptide chain [26]. Therefore, it could be speculated that the presence of Lys might not only lead to the dissociation of myosin filaments into monomers but also cause uncoiling of the coiled coil of the myosin tail via disruption of the hydrogen bonds.

#### 4.2.3. Unfolding of the Tertiary Structure

The intrinsic fluorescence of tryptophan has been commonly employed for the detection of the tertiary structural changes of proteins because tryptophan fluorescence is sensitive to the surface hydrophobicity change of protein [27,28,29]. The reduced tryptophan fluorescence and red shifting of the peak serves as an indicator of protein unfolding, which leads to the exposure of tryptophan residues to a polar environment [17]. In the present study, the presence of Lys showed significant quenching effects on the tryptophan fluorescence regardless of the ionic strength, with a slight spectral red shift (Figure 6). Therefore, it could be concluded that possible unfolding of MFP and increased interactions with the hydrophilic environment occurred due to the presence of Lys. A decreased fluorescence intensity and red shifting of peaks in myosin have been demonstrated previously in 1 mM KCl, pH 7.5 with 5 mM His [11]. Similarly, the addition of 0.05% Lys/Arg also resulted in a reduced tryptophan fluorescence intensity and a red-shifted peak position of myosin [19]. In contrast, according to the unpublished data from Takai et al. [8], neither the addition of Lys nor the NaCl concentrations affected the tryptophan fluorescence spectra of myosin. On the other hand, salt concentrations were found to influence the intrinsic fluorescence intensity of beta-lactoglobulin, which is explained by the dimerization effects of salt concentrations on the protein structure. Therefore, in the current study, the most likely explanation for the observed attenuated fluorescence intensity at the low ionic strength of 0.15 M, supported by the turbidity data, is the polymerization of MFP into filaments and the increased number of tryptophan residues buried inside the protein such that they cannot absorb the excitation light.

## 5. Conclusions

To develop low-salt meat products, it is of critical importance to increase the MFP solubility under low-salt processing conditions. The present study demonstrated that to some extent, the presence of Lys can improve MFPs’ solubility at a low ionic strength of 0.3 M NaCl. Furthermore, it could be deduced that the tertiary and secondary structural changes of MFP, including myofibril swelling, dissociation of myosin filaments, uncoiling of α-helix, and unfolding of tertiary structure, might contribute to the increased solubility of MFP at such an ionic strength. Therefore, our findings suggest the industry potential of the incorporation of Lys in product formulations to develop salt-reduced meat products. Further studies regarding the effects of Lys on MFP gelation are also warranted since the gelation behaviors of the extracted MFPs during heating are responsible for the formation of a desirable structure of low-salt meat products.

## Figures and Tables

**Figure 1 foods-11-00855-f001:**
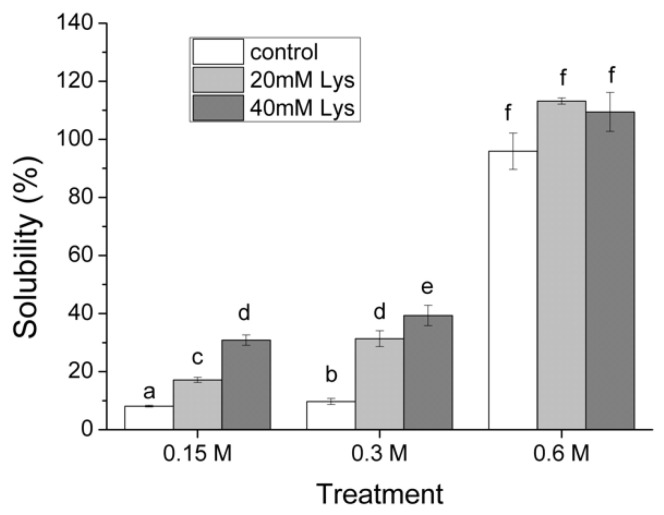
The effects of Lys on the solubility of MFP at different ionic strengths. Columns with different letters indicate significant differences at the level of *p* < 0.05. 0.15 M, treatment with ionic strength of 0.15 M NaCl; 0.3 M, treatment with ionic strength of 0.3 M NaCl; 0.6 M, treatment with ionic strength of 0.6 M NaCl; control, treatment without Lys addition; 20 mM Lys, treatment with 20 mM Lys; 40 mM Lys, treatment with 40 mM Lys.

**Figure 2 foods-11-00855-f002:**
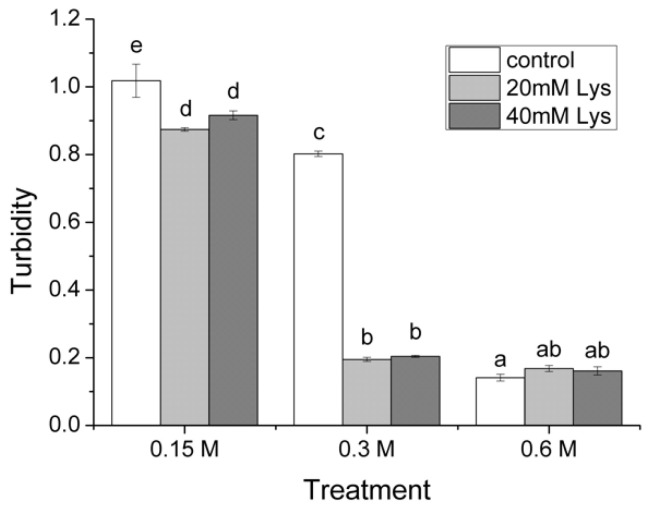
The effects of Lys on the turbidity of MFP solutions at different ionic strengths. Columns with different letters indicate significant differences at the level of *p* < 0.05. 0.15 M, treatment with ionic strength of 0.15 M NaCl; 0.3 M, treatment with ionic strength of 0.3 M NaCl; 0.6 M, treatment with ionic strength of 0.6 M NaCl; control, treatment without Lys addition; 20 mM Lys, treatment with 20 mM Lys; 40 mM Lys, treatment with 40 mM Lys.

**Figure 3 foods-11-00855-f003:**
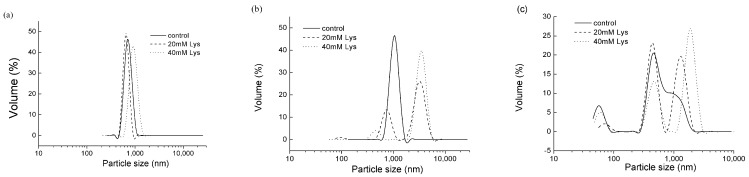
The effects of Lys on the particle size of MFP solutions at different ionic strengths. (**a**) The ionic strength of 0.15 M; (**b**) the ionic strength of 0.3 M; (**c**) the ionic strength of 0.6 M. Control, treatment without Lys addition; 20 mM Lys, treatment with 20 mM Lys; 40 mM Lys, treatment with 40 mM Lys.

**Figure 4 foods-11-00855-f004:**
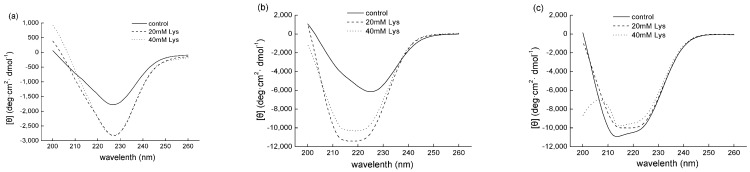
The effects of Lys on the CD spectra of MFP solutions at different ionic strengths. (**a**) The ionic strength of 0.15 M; (**b**) the ionic strength of 0.3 M; (**c**) the ionic strength of 0.6 M. control, treatment without Lys addition; 20 mM Lys, treatment with 20 mM Lys; 40 mM Lys, treatment with 40 mM Lys.

**Figure 5 foods-11-00855-f005:**
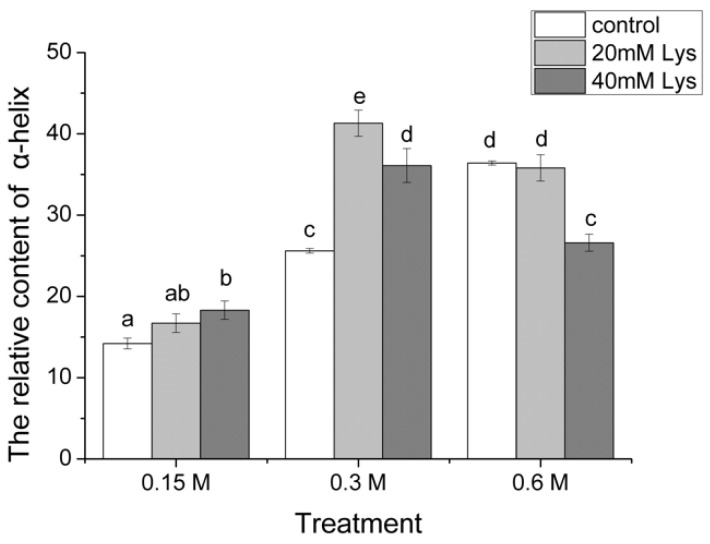
The effects of Lys on the relative content of α-helix in MFP solutions at different ionic strengths. Columns with different letters indicate significant differences at the level of *p* < 0.05. 0.15 M, treatment with ionic strength of 0.15 M NaCl; 0.3 M, treatment with ionic strength of 0.3 M NaCl; 0.6 M, treatment with ionic strength of 0.6 M NaCl; control, treatment without Lys addition; 20 mM Lys, treatment with 20 mM Lys; 40 mM Lys, treatment with 40 mM Lys.

**Figure 6 foods-11-00855-f006:**
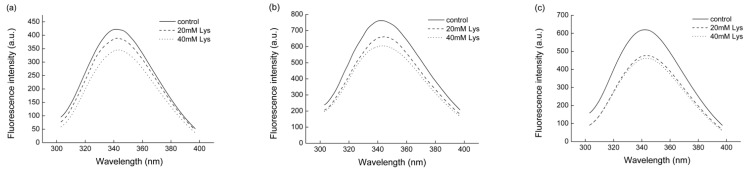
The effects of Lys on the intrinsic tryptophan fluorescence spectra of MFP solutions at different ionic strengths. (**a**) The ionic strength of 0.15 M; (**b**), the ionic strength of 0.3 M; (**c**), the ionic strength of 0.6 M. Control, treatment without Lys addition; 20 mM Lys, treatment with 20 mM Lys; 40 mM Lys, treatment with 40 mM Lys.

## Data Availability

The data presented in this study are available on request from the corresponding author.

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
