# Peer review of "The Solubility and Structures of Porcine Myofibrillar Proteins under Low-Salt Processing Conditions as Affected by the Presence of L-Lysine"

_foods, 2022, doi:10.3390/foods11060855_

Round 1
Reviewer 1 Report
Comments to authors
The study is well designed, but not new. Many studies are available which have already proven the amino acids improve functionality of the protein. This study has some scientific relevance and industrial use.
In section 2.3. for preparation of MFP suspension, pH 6 was maintained instead of 7 why ?? Could you please explain ?
In many methods authors used a term “with slight modifications” is it possible to incorporate in the methods ?
Bar diagrams for solubility and other parameters are highly confusing as two columns are white which one belong to which treatment, very difficult to know?
For swelling properties- particle size was determined. However, viscosity also would have been also measured.
Some amino acid imparts sweetness/bitterness. Lysine is also thought to be slightly bitter. There are many studies of addition of amino acids in improving functionality of the protein but sensory taste is performed seldomly. Especially taste ? Same is missing in this study also. Because, ultimately acceptability of the products is important than any other technological interventions we do. eve tough addition of lysine improves functionality.
Conclusion part may be more critical and brief
In reference section please check ref. no 7, 11, 21. FOOD ?

Reviewer 2 Report
The article is about the effect of L-lysine and ionic strength of extraction buffer against porcine myofibrillar proteins. The authors have explained the importance of low-salt processing condition from the health perspective without compromising the crucial factors contributing to palatability and sensory perception. The authors have also presented and discussed the results briefly regarding to solubilization and structural changes (swelling of myofibrils, dissociation of myosin, and unfolding of tertiary structure). However, there some correction should be addressed by the authors to make the article perfect. The comments are as follows:
- Page 2, line 84: "... other common chemicals phased ..." Please mention all the chemicals to make the sentence clear.
- Page 2, line 89: "... small cubes, ..." Please specify the size of cube.
- Page 2, line 94: "... then homogenized ..." How much the minced meat should be homogenized?
- Page 2, line 94: "... four volumes ..." The word "four volume" is unclear. Is it refer to replicate or volume of extraction buffer? The amount of extraction buffer is also unclear.
- Page 3, lines 112 - 113: "Protein concentrations of assorted MFP suspensions were adjusted to the same level of 10 mg/ml. Because the presence of Lys could lead to the pH increase of MFP solutions, which might facilitate MFP solubilization." These two sentences are quite confusing. Please rephrase the two sentences.
- Page 4, line 157: "... performed using R (version 157 4.1.2, R Foundation for Statistical Computing, Vienna, Austria) ..." The authors are encourage to share the R script.
- Page 4, line 177: Figure 1. It is difficult to differentiate between control and 20mM Lys, as the legends using same color. It is difficult to understand the result due to the legends issue. Please address the same issue on Figure 2 and 5.
